# Direct Disk Diffusion Testing and Antimicrobial Stewardship for Gram-Negative Bacteremia in the Context of High Multidrug Resistance

**DOI:** 10.3390/antibiotics14070726

**Published:** 2025-07-19

**Authors:** Wantin Sribenjalux, Pawarit Kulwongroj, Waewta Kuwatjanakul, Lumyai Wonglakorn, Kanuengnit Srisak, Natapong Manomaiwong, Atibordee Meesing

**Affiliations:** 1Division of Infectious Diseases and Tropical Medicine, Department of Internal Medicine, Faculty of Medicine, Khon Kaen University, Khon Kaen 40002, Thailand; natapong.man@hotmail.com (N.M.); atibordee@kku.ac.th (A.M.); 2Research and Diagnostic Center for Emerging Infectious Diseases (RCEID), Khon Kaen University, Khon Kaen 40002, Thailand; 3Department of Internal Medicine, Faculty of Medicine, Khon Kaen University, Khon Kaen 40002, Thailand; pavara@kku.ac.th; 4Microbiology Unit, Clinical Laboratory Section, Srinagarind Hospital Faculty of Medicine, Khon Kaen University, Khon Kaen 40002, Thailand; pwaewt@kku.ac.th (W.K.); wlumya@kku.ac.th (L.W.); kanusr@kku.ac.th (K.S.)

**Keywords:** disk diffusion, rapid diagnostics, antimicrobial stewardship, bacteremia, Gram-negative

## Abstract

**Background:** Combining direct disk diffusion (DD) testing with antimicrobial stewardship (AMS) may optimize antibiotic use and improve outcomes in patients with Gram-negative bloodstream infections (GNBSIs). **Methods:** This quasi-experimental study was conducted at Srinagarind Hospital, Khon Kaen University, between 13 September 2022 and 11 April 2023. Patients with GNBSIs were enrolled during two phases: a standard care phase (13 September 2022–2 January 2023) and an intervention phase (16 January 2023–11 April 2023), during which therapy adjustments were guided by DD results interpreted by infectious disease specialists. **Results:** Among the 141 patients included (68 in the standard care group and 73 in the intervention group), the mean age was 61.7 years, and 60.2% were male. *Escherichia coli* (36.5%) and *Klebsiella pneumoniae* (27.6%) were the most frequently isolated pathogens, with intra-abdominal and urinary tract infections being the most common sources. Multidrug-resistant (MDR) organisms were identified in 48.9% of cases. Compared to standard care, the intervention group had a significantly shorter median time to optimal therapy (40.0 vs. 59.1 h, *p* = 0.037) and a higher proportion of patients receiving optimal therapy within 72 h (86.2% vs. 62.3%, *p* = 0.002). While 30-day mortality did not differ significantly between groups (17.2% vs. 16.7%, *p* = 0.98), MDR bacteremia and ICU admission were associated with increased mortality. In contrast, receiving optimal therapy within 72 h was associated with reduced mortality. **Conclusion:** Direct DD testing combined with AMS significantly reduced the time to optimal antibiotic therapy and decreased inappropriate antibiotic use in GNBSI patients. Achieving optimal therapy within 72 h was associated with a trend toward reduced mortality.

## 1. Introduction

Gram-negative blood stream infections (GNBSIs) are a major cause of morbidity and mortality in hospitalized patients, with 30-day mortality rates reaching up to 35% [1]. Inappropriate empirical antibiotic therapy, whether due to inadequate pathogen coverage or unnecessary broad-spectrum antibiotic use, is associated with increased mortality and healthcare costs [2,3,4,5]. A meta-analysis reported that 14.1% to 78.9% of critically ill patients receive inappropriate empirical antibiotic therapy [5]. In Thailand, multidrug-resistant (MDR) Gram-negative bacteria are commonly encountered, including in rural community hospitals, where they accounted for 30.8% of Gram-negative infections [6]. In university hospitals, the prevalence rose to 48.8%, with 56.7% of affected patients receiving inappropriate empirical antibiotic therapy [7].

The timely administration of appropriate antibiotics is essential to improving outcomes for patients with sepsis and septic shock, as emphasized in as emphasized in the 2021 Surviving Sepsis Campaign guidelines [8]. However, the increasing prevalence of MDR and extremely drug-resistant (XDR) pathogens complicates the selection of effective empirical therapy. In Thailand, carbapenems are the preferred empirical treatment for infections caused by extended-spectrum beta-lactamase (ESBL)-producing pathogens, following guidelines from the European Society of Clinical Microbiology and Infectious Diseases (ESCMID) and the Infectious Diseases Society of America (IDSA) [9,10]. For carbapenem-resistant organisms, however, many novel agents recommended by the IDSA, such as sulbactam/durlobactam, cefiderocol, and aztreonam, are currently unavailable in the country. Moreover, in carbapenem-resistant Enterobacterales (CRE), the predominant resistance mechanism is metallo-beta-lactamase production, which limits the effectiveness of newer beta-lactam/beta-lactamase inhibitor combinations unless used with aztreonam [11]. Consequently, empirical treatment often relies on combinations of older antibiotics, including colistin, fosfomycin, tigecycline, aminoglycosides, and high-dose carbapenems, in accordance with ESCMID’s alternative regimens. The choice of agents is typically guided by local susceptibility patterns and the site of infection. However, the potential toxicity of these agents often makes clinicians reluctant to initiate therapy without confirmed susceptibility data.

To address this challenge, efforts have been made to accelerate susceptibility testing. Various rapid antimicrobial susceptibility testing (RAST) techniques, such as matrix-assisted laser desorption ionization time-of-flight mass spectrometry (MALDI-TOF MS), fluorescent in situ hybridization (FISH), molecular detection systems based on nucleic acid amplification tests (NAATs), and microarrays, can provide susceptibility results within 8 h of a positive blood culture, compared to the 48 to 96 h required by conventional methods, without compromising accuracy [12,13,14,15]. Integrating RAST into antimicrobial stewardship (AMS) programs can significantly shorten the time to optimal antibiotic therapy, with reported reductions ranging from 6.3 to 34.6 h. However, previous studies have not demonstrated a clear mortality benefit and have shown inconsistent reductions in length of stay (LOS), possibly due to variations in study populations, resistance patterns, and healthcare settings [16,17,18,19]. In addition, the high cost and need for specialized equipment and expertise limit the implementation of many RAST techniques in resource-limited settings.

The direct disk diffusion (DD) is a simpler, more accessible form of RAST that is performed directly from positive blood culture broth, without the need to isolate pure bacterial colonies. It is easy to implement, has a lower cost per test compared to other RAST methods, and provides acceptable accuracy [20,21,22]. The Clinical and Laboratory Standards Institute (CLSI) now endorses direct DD as a validated method for Enterobacterales, *Pseudomonas aeruginosa*, and *Acinetobacter* spp., allowing for earlier targeted therapy adjustments [23]. Few studies have examined the clinical impact of combining direct DD testing with AMS in patients with GNBSIs, particularly in settings with a high burden of MDR organisms [24,25,26]. Given the high prevalence of MDR Gram-negative pathogens in regions like Thailand, the potential benefits of integrating RAST with AMS may be even greater than those observed in studies from areas with lower resistance rates [27].

This study aimed to evaluate whether integrating direct DD with AMS could shorten the time to optimal antibiotic therapy in patients with GNBSI. A secondary objective was to determine whether this approach could also reduce in-hospital mortality.

## 2. Results

### 2.1. Patient Characteristics

A total of 141 patients were included in the study, with 68 in the standard of care (SOC) group and 73 in the intervention group (Figure 1). The average age of the overall study population was approximately 60 years, and the majority of patients were male. Over half were admitted to the medical ward followed by the surgery unit. (Table 1). The most prevalent comorbidities in the SOC group were solid organ malignancy (33.8%), diabetes mellitus (27.9%), and renal disease (17.6%). In contrast, diabetes mellitus was the most common comorbidity in the intervention group (32.9%), followed by solid organ malignancy (28.8%) and renal disease (19.2%). However, there were no statistically significant differences between the two groups.

On the day of blood culture collection, clinical and laboratory parameters were largely similar between the two groups (Table 2). There were no significant differences in the Pitt bacteremia score or Sequential Organ Failure Assessment (SOFA) score. However, WBC count was significantly higher in the intervention group (11,809 ± 7800 vs. 15,442 ± 9893 cells/mm^3^; *p* = 0.016), whereas serum albumin was lower in the SOC group (3.2 ± 0.9 vs. 3.4 ± 1.0 g/dL; *p* = 0.049).

### 2.2. Infectious Source, Microbiological Findings, and Antimicrobial Therapy

The primary sources of infection in both groups were intra-abdominal infections (33.8% in the SOC group vs. 26.0% in the intervention group, *p* = 0.396) and urinary tract infections (20.6% vs. 31.5%, *p* = 0.396) (Table 3). The prevalence of bacteremia caused by MDR pathogens was similar between the SOC and intervention groups (50.0% vs. 47.9%, *p* = 0.807). Carbapenem-resistant bacteremia was observed in both groups (35.5% in the SOC group vs. 24.7% in the intervention group, *p* = 0.168), with no statistically significant difference.

The most frequently isolated pathogens were *Escherichia coli* (36.5%) and *Klebsiella pneumoniae* (27.6%), both of which commonly reside in the gastrointestinal tract and correlate with the primary sources of infection in these patients. These were followed by *Acinetobacter baumannii* (11.5%) and *Pseudomonas* spp. (5.1%) (Figure 2).

### 2.3. Empirical and Definitive Antibiotic Therapy

Empirical antibiotics included meropenem (36.2%), piperacillin/tazobactam (15.4%), ceftriaxone (11.2%), and colistin (8.5%), while definitive antibiotic therapies consisted of ceftriaxone (24.6%), meropenem (24.0%), colistin (15.1%), and cefoperazone/sulbactam (5.0%). Detail of empirical and definite antibiotic regimens were shown as Appendix A.

### 2.4. Accuracy of Direct Disk Diffusion Test

The overall categorical agreement (CA) between RAST and standard culture-based susceptibility testing, based on retrospective interpretation using the 2025 CLSI clinical breakpoints, was 94.12%. Minor errors (mE), major errors (ME), and very major errors (VME) occurred at rates of 3.68%, 1.47%, and 0.74%, respectively. A detailed comparison between RAST and standard susceptibility testing is provided in the Appendix A.

### 2.5. Antibiotic Adjustments and Time to Optimal Therapy

Among patients in the intervention group, 38.4% underwent escalation, 27.4% underwent de-escalation, and 34.2% had no change in therapy following RAST-guided recommendations (Table 4).

The median time to optimal targeted antibiotic therapy was significantly reduced in the intervention group compared to the standard care group (40.0 vs. 59.1 h, *p* = 0.037). A significantly higher proportion of patients in the intervention group achieved optimal antibiotic therapy within 48 h (68.5% vs. 36.8%, *p* < 0.001) and within 72 h (86.2% vs. 62.3%, *p* = 0.002) (Table 4). Furthermore, the rate of ineffective antibiotic use was significantly lower in the intervention group at both 48 h (12.3% vs. 27.9%, *p* = 0.002) and 72 h (2.7% vs. 17.6%, *p* = 0.003) (Table 4).

### 2.6. Clinical Outcomes

There were no significant differences between the intervention and standard care groups in terms of intensive care unit (ICU) admission within 72 h after antimicrobial susceptibility testing (AST) results (20.5% vs. 22.1%, *p* = 0.872), duration of vasopressor use (80.5 vs. 51.0 h, *p* = 0.208), SOFA score at day 3 (median: 4 vs. 4, *p* = 0.740), or hospital length of stay (11.0 vs. 13.0 days, *p* = 0.195). Similarly, the 30-day mortality rate was comparable between the intervention and standard care groups (17.2% vs. 16.7%, *p* = 0.980). The median hospitalization cost was lower in the intervention group (98,606 vs. 145,230 Baht), but the difference was not statistically significant (*p* = 0.152)

### 2.7. Subgroup Analysis in the Patients with Multidrug-Resistant Bacteremia

Among 67 patients with MDR bacteremia—32 in the standard care group and 35 in the intervention group—those in the intervention group were more likely to receive appropriate antibiotics. Within 48 h, 50.0% of patients in the standard care group received optimal antibiotics compared to 74.3% in the intervention group (*p* = 0.040). Within 72 h, the corresponding rates were 68.8% vs. 91.4% (*p* = 0.019). There were no deaths within 72 h of enrollment in either group. ICU admission within 72 h occurred in 25.0% of patients in the standard care group and 28.6% in the intervention group (*p* = 0.742). The 30-day mortality rate was also similar between the groups (25.0% vs. 25.7%, *p* = 0.946).

### 2.8. Factors Associated with Death

In univariate analysis, factors associated with increased mortality included bacteremia caused by an MDR pathogen [crude odds ratio (cOR) with 95% confidence intervals (CI) = 3.04 (1.23–7.50), *p* = 0.016], the requirement for mechanical ventilator support [cOR = 3.14 (1.24–7.95), *p* = 0.016], ICU admission [cOR = 6.30 (2.59–12.7), *p* < 0.001], pneumonia as the primary source of infection [cOR = 4.20 (1.61–11.0), *p* = 0.003], and a body mass index (BMI) < 21 kg/m^2^ [cOR = 2.43 (1.04–5.71), *p* = 0.041]. Among all factors analyzed, only the achievement of optimal antibiotic therapy within 72 h was significantly associated with reduced mortality [cOR = 0.39 (0.16–0.95), *p* = 0.037], highlighting its potential impact on patient outcomes (Table 5).

Multivariate analysis confirmed that bacteremia caused by an MDR pathogen [adjusted odds ratio (aOR) with 95%CI = 3.01 (1.02–8.87), *p* = 0.046], ICU admission [aOR = 6.13 (2.03–18.5), *p* = 0.001], and BMI <21 kg/m^2^ [aOR = 4.58 (1.54–13.7), *p* = 0.006] remained independent predictors of mortality. Although achieving optimal antibiotic therapy within 72 h showed a protective trend, it did not reach statistical significance [aOR = 0.42 (0.15–1.21), *p* = 0.108] (Table 5). The need for mechanical ventilation was excluded from the multivariate analysis to avoid collinearity with ICU admission, which demonstrated a stronger association based on the odds ratio and *p*-Value.

## 3. Discussion

This study demonstrated that integrating direct DD testing with AMS significantly reduced the time to optimal antibiotic therapy and decreased ineffective antibiotic use in patients with GNBSI. However, no significant difference in 30-day mortality was observed between the standard care and intervention groups. These findings align with previous studies indicating that, while RAST-driven interventions improve antimicrobial optimization, they do not always translate into mortality benefits [17,18,26]. The intervention significantly shortened the median time to optimal antibiotic therapy, consistent with previous studies reporting reductions of 6.2 to 34.6 h when RAST is used alongside AMS [16,17,18,19]. Given the high prevalence of MDR Gram-negative pathogens in Thailand, early therapy optimization is critical, especially in settings where empirical therapy may fail to cover resistant organisms [7,28]. The observed decrease in ineffective antibiotic use at 48 and 72 h highlights the clinical value of RAST in guiding rapid antimicrobial adjustments, ultimately reducing unnecessary broad-spectrum antibiotic exposure.

Although the intervention successfully optimized antibiotic therapy, no significant difference in 30-day mortality was observed between the two groups. Several factors may account for this finding. Previous studies, particularly those with a lower proportion of patients infected with MDR organisms, have suggested that empirical antibiotic therapy is often appropriate from the outset [17]. In our study, despite a higher proportion of patients with MDR Gram-negative bacteremia, approximately two-thirds of antibiotic adjustments were either de-escalations or no changes at all. This suggests that most empirical regimens were already suitable, thereby limiting the potential impact of the intervention on mortality. Furthermore, consistent with prior studies that did not report significant effects on mortality [17,18,24,25,26,27], it is important to recognize that outcomes in bloodstream infections are influenced by multiple factors beyond the timing and choice of antibiotics. These include the patient’s baseline severity of illness, comorbidities, adequacy of source control, and immune status. In our analysis, ICU admission and MDR bacteremia were identified as independent predictors of mortality, highlighting the predominant role of disease severity and resistance patterns in determining patient outcomes. In addition, our study may have been underpowered to detect a statistically significant difference in mortality, as the sample size was calculated based on the primary outcome—time to optimal antibiotic therapy—rather than mortality.

The impact of combining RAST with AMS can vary depending on how comprehensive the AMS strategy is. AMS refers to coordinated interventions designed to improve and measure the appropriate use of antimicrobial agents. This includes optimizing drug selection, dosing, route, and duration of therapy. In our study, the AMS intervention focused only on recommending appropriate empirical therapy and did not extend to guidance on definitive treatment, treatment duration, or oral step-down strategies. In contrast, Ventres JJ et al. demonstrated that when AMS recommendations also included treatment duration and oral step-down, the intervention led to reduced hospital length of stay and readmission rates [19]. This suggests that broader AMS involvement may contribute to improved clinical outcomes.

In this study, direct DD testing demonstrated a high categorical agreement of 94.12% compared to standard culture-based susceptibility testing, with only one VME observed. This supports its reliability in clinical decision-making. These results are consistent with previous studies validating direct DD as an accurate and reproducible method for Enterobacterales, *P. aeruginosa*, and *Acinetobacter* spp., as endorsed by CLSI guidelines [20,21,22,23]. In addition, we extended direct DD testing to other antibiotics, including ceftazidime/avibactam, cefoperazone/sulbactam, fosfomycin, and sitafloxacin. Although further studies are needed, these agents warrant attention due to rising antimicrobial resistance and their potential future importance. Given the critical nature of bloodstream infections, ensuring rapid and accurate antimicrobial susceptibility results is essential to facilitating earlier targeted therapy adjustments.

This study has several limitations. First, being a single-center study with a small sample size, the findings may have limited generalizability to other healthcare settings with different resistance patterns. Additionally, the small sample size may have reduced the statistical power for analyzing secondary outcomes, including mortality. Second, the quasi-experimental design may have introduced differences in baseline characteristics, such as the slightly elevated baseline WBC in the SOC group, which could have contributed to worse outcomes. Furthermore, the presence of unmeasured confounding variables cannot be ruled out. Nonetheless, the majority of baseline characteristics were comparable between the two groups. Third, due to technical constraints, we used cefotaxime and imipenem disks instead of ceftriaxone and meropenem, respectively. According to CLSI guidelines, the disk content and inhibition zone diameters for cefotaxime and ceftriaxone, as well as imipenem and meropenem, are similar [23]. Therefore, this substitution is unlikely to have meaningfully affected the study outcomes. Finally, a considerable number of cases in the intervention phase were excluded due to primary physicians refusing to participate for various reasons. Most of these were critically ill patients, for whom physicians were hesitant to de-escalate antibiotics based on preliminary results. Including these cases might have influenced the study outcomes, potentially showing a greater reduction in treatment costs. Future multicenter randomized controlled trials with larger sample sizes, calculated based on mortality rates, are needed to more definitively assess the clinical impact of RAST-integrated AMS programs, including their effects on long-term outcomes, cost-effectiveness, and resistance patterns.

## 4. Materials and Methods

### 4.1. Study Design

This quasi-experimental study was conducted at Srinagarind Hospital, a university hospital in Northeast Thailand, between 13 September 2022, and 11 April 2023. The study consisted of two phases: a standard care phase and an intervention phase. During the standard care phase, which took place between 13 September 2022, and 2 January 2023, patients received routine clinical management without RAST-guided antimicrobial interventions. The process began after a blood culture turned positive. The laboratory notified the ward where the patient was admitted to alert the primary physician. Meanwhile, subcultures from the broth were performed on blood agar and Mueller-Hinton agar (MHA) according to CLSI protocol. After 10 to 18 h, once a pure colony was isolated, MALDI-TOF MS was used to identify the genus and species of the microorganism, which required approximately 4 h before the results were reported in the hospital system. Antimicrobial susceptibility testing was then performed using automated broth microdilution (Sensititre ARIS 2X, Thermo Scientific, Lenexa, KS, USA), which took an additional 20 h to complete before results became available. The intervention phase took place between 16 January 2023 and 11 April 2023. In addition to standard procedures, direct disk diffusion (DD) testing was performed using broth from positive blood culture bottles, as described in a later section. Susceptibility interpretation based on zone diameter was made after bacterial identification by MALDI-TOF MS. Antimicrobial therapy was then adjusted with input from infectious disease specialists.

### 4.2. Study Population

Patients were eligible for inclusion if they were 18 years of age or older and had a confirmed GNBSI, as determined by a positive blood culture. Patients were excluded if they died within 24 h after blood culture collection or if their primary physician did not follow the antimicrobial recommendations provided by the research team.

The sample size was calculated using the n4studies tool, based on the formula shown in Figure 3. In this formula, P1 and P2 represent the proportions of patients who received appropriate antibiotics in the intervention and control groups, respectively [17]. The calculation determined that at least 68 patients per group were needed to achieve 80% statistical power. During the standard care phase, 71 patients were enrolled, but three patients were excluded due to participation in other clinical studies that involved antimicrobial modifications, resulting in 68 patients included in the final analysis. During the intervention phase, 113 patients were screened, but 40 were excluded due to non-adherence to antimicrobial recommendations by primary physicians, leaving 73 patients in the final analysis (Figure 1).

### 4.3. Disk Diffusion Susceptibility Testing

In the intervention group, RAST was performed using direct DD testing on positive blood culture bottles. The procedure involved mixing the blood culture bottle by inverting it 5–10 times to ensure homogeneity. A small volume of the culture suspension was then aspirated and dispensed onto a MHA plate in four separate drops. The suspension was evenly spread across the agar surface before placing antibiotic-impregnated disks onto the plate. The MHA plates were incubated under ambient aerobic conditions at 35 ± 2 °C for 8–10 h. After incubation, the diameters of the inhibition zones were measured, and the results were interpreted according to the most recent CLSI guidelines available at the time of the study (2022) [29].

In addition to antibiotics recommended by the CLSI guidelines, we also evaluated other available agents, such as ceftazidime/avibactam, cefoperazone/sulbactam, fosfomycin, imipenem, and off-label options like sitafloxacin, to explore potential treatment options for patients with XDR organisms. In Thailand, sitafloxacin has been used as an oral step-down option for infections caused by XDR organisms, particularly when co-trimoxazole and other fluoroquinolones, such as ciprofloxacin and levofloxacin, are commonly resistant [30]. These preliminary findings supported the AMS team in guiding early antimicrobial optimization before the final culture and susceptibility results became available.

### 4.4. Intervention and Antimicrobial Stewardship Protocol

For patients in the intervention group, an RAST-guided antimicrobial optimization protocol was implemented. When Gram-negative bacilli were detected in a positive blood culture, the microbiology laboratory performed direct DD testing to determine preliminary pathogen identification and antimicrobial susceptibility results. The AMS team, which included infectious disease specialists, reviewed the laboratory results along with the patient’s clinical condition. Therapy recommendations were then made based on international treatment guidelines, with modifications tailored to antibiotic availability in Thailand. The research team contacted the primary physician to communicate these recommendations, and if accepted, the antimicrobial regimen was adjusted accordingly.

The antimicrobial selection process was guided by standard treatment guidelines and the healthcare entitlements available in Thailand, including the Universal Coverage Scheme (UCS), Social Security Scheme (SSS), and Civil Servant Medical Benefit Scheme (CSMBS). The selection also incorporated pharmacokinetic and pharmacodynamic principles, while considering the site of infection and the bacterial resistance profile. In cases of ESBL-producing pathogens, carbapenems were used as the preferred treatment. For AmpC β-lactamase-producing Enterobacterales (AmpC-E), carbapenems were also selected. In patients with carbapenem-resistant Enterobacterales (CRE) infections, combination therapy with colistin and a susceptible agent was administered. Infections caused by carbapenem-resistant *Acinetobacter baumannii* (CRAB) were treated with colistin in combination with sulbactam [31,32].

### 4.5. Definitions of Study Outcomes

Optimal targeted antibiotic therapy defined as the administration of the narrowest-spectrum antibiotic to which the causative pathogen was susceptible. The selected agent must have appropriate pharmacokinetics and pharmacodynamics (PK/PD) for the primary infection site, or there should be no alternative narrower-spectrum agent available.

Ineffective antibiotic therapy defined as receiving an antibiotic to which the causative pathogen was resistant or had intermediate susceptibility, except for colistin, which has no established susceptibility breakpoint. This category also included cases where patients did not receive appropriate antibiotic therapy for their infection or received an agent with inappropriate PK/PD for the primary infection site, despite the availability of a more suitable alternative [26].

### 4.6. Outcome Measures

The primary outcome was the time to optimal antibiotic therapy, defined as the time from blood culture collection to the administration of the optimal targeted antibiotic based on susceptibility results.

Secondary outcomes included the proportion of patients achieving optimal therapy within 48 and 72 h, the rate of ineffective antibiotic use at 48 and 72 h, and the incidence of ICU admission within 72 h after AST results were officially reported. Additional secondary outcomes included the incidence of hospital-onset *Clostridioides difficile* infection, the duration of vasopressor use, the SOFA score on day 3, the hospital LOS, the proportion of patients discharged alive, and 30-day mortality along with associated risk factors.

### 4.7. Statistical Analysis

Categorical variables were reported as frequencies and percentages and compared using the chi-squared or Fisher’s exact test. Continuous variables were analyzed using the independent *t*-test or the Mann-Whitney U test, depending on data distribution. Results are presented as mean with standard deviation (SD) or median with interquartile range (IQR). Mean ± SD was used for normally distributed data, while the median was preferred for data with outliers, as it is less affected by extreme values and better represents the central tendency in such cases. Time to optimal antibiotic therapy, the primary outcome, was compared between groups using the Mann–Whitney U test due to non-normal distribution. The proportions of patients achieving optimal therapy or receiving ineffective therapy at 48 and 72 h were compared using the chi-squared test.

For mortality analysis, univariate logistic regression was performed to evaluate the association between clinical factors and 30-day mortality. Variables with *p* < 0.10 in the univariate analysis were included in a multivariate logistic regression model to identify independent predictors of mortality. aOR with 95%CI were reported. Statistical significance was defined as *p* < 0.05. All analyses were performed using SPSS version 28.0.0.0

### 4.8. Ethical Consideration

This study was approved by the Institutional Review Board of Khon Kaen University (Reference Number: HE651356), and informed consent was obtained from all participants prior to enrollment.

## 5. Conclusions

The combination of RAST and AMS significantly improved the time to optimal antibiotic therapy and reduced ineffective antibiotic use in patients with GNBSI. Although no significant mortality benefit was observed, the protective trend in univariate analysis suggests that earlier targeted therapy may contribute to improved outcomes. Given the high burden of MDR infections, particularly in Thailand, implementing RAST-driven AMS programs may be a useful strategy to support antimicrobial optimization. However, given the study’s limitations, further large-scale, multicenter randomized trials are needed to confirm the clinical impact, particularly on mortality.

## Figures and Tables

**Figure 1 antibiotics-14-00726-f001:**
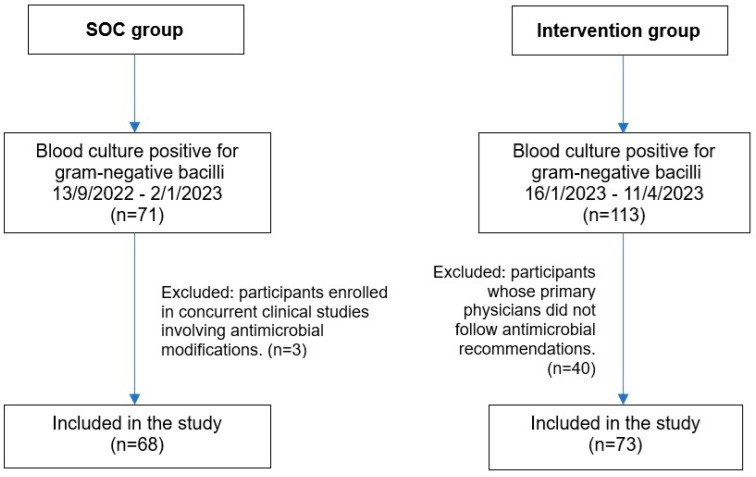
Flow diagram of patient enrollment and inclusion in the SOC and intervention groups. Abbreviation: SOC, standard of care.

**Figure 2 antibiotics-14-00726-f002:**
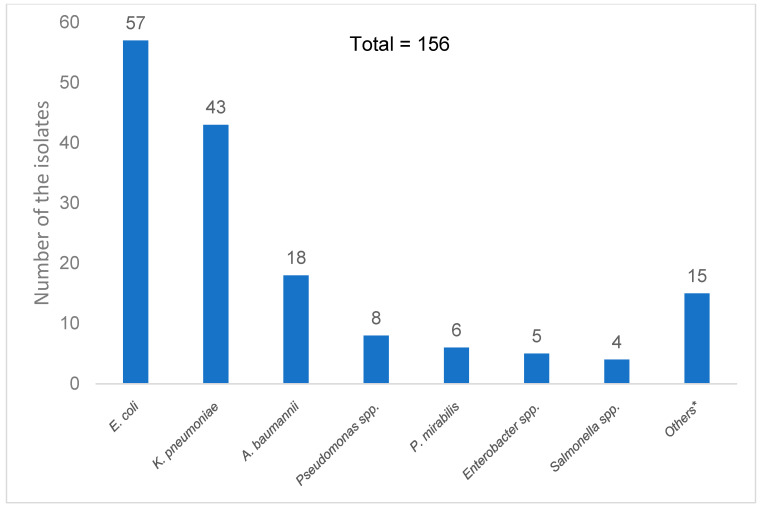
Distribution of Gram-negative bacilli species isolated from the blood cultures of all participants. * Others are *Serratia marcescens*, *Stenotrophomonas maltophilla*, *Aeromonas hydrophila*, *Aeromonas caviae*, *Providencia stuartii*, *Pantoea* spp., *Providencia stuartii*, *Pluralibacter gergoviae*, *Klebsiella aerogenes*, *Klebsiella oxytoca*, *Acinetobacter baylyi*, *Burkholderia pseudomallei*, and *Burkholderia cepacia* complex.

**Figure 3 antibiotics-14-00726-f003:**
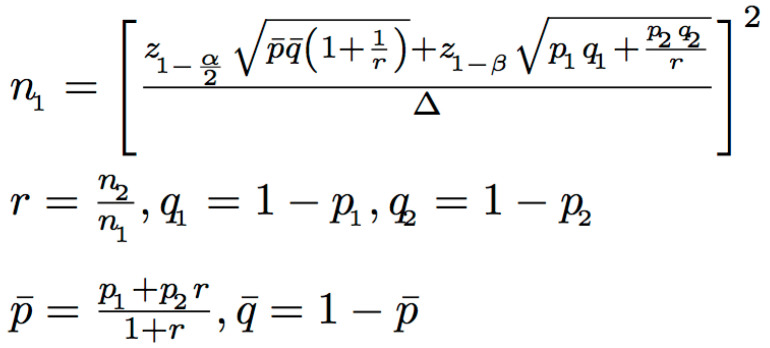
Formula used to calculate the required sample size based on study parameters.

**Table 1 antibiotics-14-00726-t001:** Baseline characteristics of patients in the SOC and intervention groups.

Characteristics		SOC (N = 68)	Intervention (N = 73)	*p*-Value
Male sex, no. (%)		42 (61.8)	43 (58.9)	0.729
Age (years, mean (SD))		60.4 (16.1)	62.6 (17.3)	0.293
BMI (kg/m^2^, median (IQR))		22.6 (6.4)	22.7 (6.1)	0.800
Type of admission unit, no. (%)				0.352
	Medicine	33 (48.5)	44 (60.3)	
	Surgery	29 (42.6)	23 (31.5)	
	Others *	6 (8.8)	6 (8.2)	
Charlson comorbidity index (median, (IQR))		4 (5)	4 (4)	0.695
Underlying disease, no. (%)				
	Solid organ malignancy	23 (33.8)	21 (28.8)	0.517
	DM	19 (27.9)	24 (32.9)	0.525
	Renal disease	12 (17.6)	14 (19.2)	0.815
	Liver disease	8 (11.8)	11 (15.1)	0.566
	Hematologic malignancy	8 (11.8)	9 (12.3)	0.918
	Coronary artery disease	7 (10.3)	4 (5.5)	0.287
	Cerebrovascular disease or TIA	3 (4.4)	4 (5.5)	1.000
	Solid organ transplantation	4 (5.9)	3 (4.1)	0.771
	Currently receiving immunosuppressive agent(s)	20 (29.4)	19 (26.0)	0.653

Abbreviations: DM, diabetes mellitus; IQR, interquartile range; SOC, standard of care; TIA, transient ischemic attack. * Orthopedic, obstetrics and gynecology, otorhinolaryngology, rehabilitation, and emergency short stay ward.

**Table 2 antibiotics-14-00726-t002:** Clinical and laboratory parameters on the day of blood culture collection in the SOC and intervention groups.

	SOC (N = 68)	Intervention (N = 73)	*p*-Value
Clinical parameters			
BT (Celsius degree, median (IQR))	38.7 (1.3)	38.6 (1.8)	0.856
SBP (mmHg, mean (SD))	130.3 (20.7)	131.0 (21.8)	0.859
MAP (mmHg, mean (SD))	92.0 (12.7)	92.0 (14.2)	0.890
Glasgow Coma Score (median (range))	15 (9–15)	15 (3–15)	0.707
Acute hypotensive episode (%)	34 (50.0)	31 (42.5)	0.370
Required vasopressor (%)	24 (35.3)	19 (26.0)	0.232
Required mechanical ventilation (%)	14 (20.6)	14 (19.2)	0.843
Neutropenia (%)	6 (8.8)	8 (11.0)	0.672
Admitted in ICU (%)	22 (32.4)	19 (26.0)	0.409
Pitt bacteremia score (median (range))	2 (0–7)	2 (0–8)	0.295
SOFA score (median (range))	5 (0–18)	4 (0–14)	0.100
Laboratory parameters			
Hb (g/dL, median (IQR))	9.5 (3.1)	10.0 (3.6)	0.737
WBC (cell/mm^3^, mean (SD))	**15,442 (9893)**	**11,809 (7800)**	**0.016**
PMN (%, median (IQR))	84.2 (18)	84.1 (23)	0.506
Platelets (×10^3^ cell/mm^3^, median (IQR))	202.5 (167)	176.5 (183)	0.342
Serum creatinine (mg/dL, median (IQR))	2.2 (1.4)	1.5 (0.7)	0.498
ALT (U/L, median (IQR))	50 (78)	50 (89)	0.753
TB (mg/dL, median (IQR))	1.3 (4.7)	1.1 (1.9)	0.641
Albumin (g/dL, median (IQR))	**3.2 (0.9)**	**3.4 (1.0)**	**0.049**
Lactate (mg/dL, median (IQR))	21.4 (19.9)	23.1 (28.1)	0.555
PaO_2_:FiO_2_ ratio (median (IQR))	273 (213)	350 (150)	0.076

Abbreviations: ALT, alanine transaminase; BT, body temperature; FiO_2_, fraction of inspired oxygen; Hb, hemoglobin; ICU, intensive care unit; IQR, interquartile range; MAP, mean arterial pressure; PaO_2_, partial pressure of oxygen; PMN, polymorphonuclear; SBP, systolic blood pressure; SOC, standard of care; SOFA, sequential organ failure assessment; TB, total bilirubin. Bold values indicate statistical significance.

**Table 3 antibiotics-14-00726-t003:** Infection-related clinical information and pathogen profile in the SOC and intervention groups.

Infectious Information		SOC (N = 68)	Intervention (N = 73)	*p*-Value
Time from admission to blood culture collection (hours, median [IQR])		40.0 (293.5)	21.3 (236.1)	0.998
Healthcare-associated infection (%)		35 (51.5)	44 (60.3)	0.293
Source of infection (%)				0.396
	Intra-abdominal *	23 (33.8)	19 (26.0)	
	Urinary tract	14 (20.6)	23 (31.5)	
	Pneumonia	9 (13.2)	15 (20.5)	
	Primary bacteremia	13 (19.1)	9 (12.3)	
	Catheter-related blood stream infection	5 (7.4)	3 (4.1)	
	Skin and soft tissue	4 (5.9)	4 (5.5)	
Polymicrobial Gram-negative bacteremia (%)		10 (14.7)	4 (5.5)	0.067
Bacteremia from MDR pathogen		34 (50.0)	35 (47.9)	0.807
Bacteremia from carbapenem-resistant pathogen		24 (35.3)	18 (24.7)	0.168

Abbreviations: IQR, interquartile range; MDR, multidrug-resistant, SOC, standard of care; * including acute cholangitis, acute cholecystitis, liver abscess, infected pancreatic necrosis, infectious diarrhea, spontaneous bacterial peritonitis.

**Table 4 antibiotics-14-00726-t004:** Comparison of clinical outcomes between SOC and RAST-guided antibiotic management.

Outcomes	Standard of Care (N = 68)	Intervention (N = 73)	*p*-Value
Antibiotic adjustments guided by RAST (%) Escalation De-escalation No adjustment	0 (0.0) 0 (0.0) 0 (0.0)	28 (38.4) 20 (27.4) 25 (34.2)	NA
Time to optimal targeted antibiotic (hour, median (IQR))	**59.1 (68.1)**	**40.0 (35.0)**	**0.037**
Optimal targeted antibiotic in 48 h (%)	**25 (36.8)**	**50 (68.5)**	**<0.001**
Optimal targeted antibiotic in 72 h (%)	**43 (62.3)**	**63 (86.2)**	**0.002**
Ineffective antibiotic in 48 h (%)	**19 (27.9)**	**9 (12.3)**	**0.002**
Ineffective antibiotic in 72 h (%)	**12 (17.6)**	**2 (2.7)**	**0.003**
Death within 72 h after official AST reporting (%)	0 (0.0)	0 (0.0)	NA
In the ICU 72 h after standard AST reporting (%)	15 (22.1)	15 (20.5)	0.872
Hospital-onset CDI (%)	5 (7.4)	4 (5.5)	0.738
Duration of vasopressor (hour, median (IQR))	51.0 (68.3)	80.5 (115.8)	0.208
SOFA score at day 3 (median (IQR))	4 (5)	4 (6)	0.740
Hospital LOS (day, median (IQR))	13 (11.8)	11 (11.0)	0.195
Discharge alive (%)	56 (82.4)	58 (79.5)	0.662
30-day mortality (%)	12 (16.7)	13 (17.2)	0.980
Cost in admission (Baht, median (IQR))	145,230 (226,433)	98,606 (249,940)	0.152

Abbreviations: AST, antimicrobial susceptibility testing; CDI, *Clostridioides difficile* infection; ICU, intensive care unit; IQR, interquartile range; LOS, length of stay; NA, not applicable; RAST, rapid antimicrobial susceptibility testing; SOFA, sequential organ failure assessment. Bold values indicate statistical significance.

**Table 5 antibiotics-14-00726-t005:** Factors associated with in-hospital mortality or discharge in critical condition among patients with Gram-negative septicemia: univariate and multivariate analysis.

Factors	cOR (95%CI)	*p*-Value	aOR (95%CI)	*p*-Value
Optimal targeted antibiotic in 72 h	0.39 (0.16–0.95)	0.037	0.42 (0.15–1.21)	0.108
Application of direct DD testing	1.21 (0.52–2.80)	0.662	…	…
Bacteremia from MDR pathogen	**3.04 (1.23–7.50)**	**0.016**	**3.01 (1.02–8.87)**	**0.046**
Required mechanical ventilator support	3.14 (1.24–7.95)	0.016	…	…
Admitted in ICU	**6.30 (2.59–12.7)**	**<0.001**	**6.13 (2.03–18.5)**	**0.001**
Pneumonia	4.20 (1.61–11.0)	0.003	1.86 (0.60–5.77)	0.286
Charlson comorbidity index ≥5	1.85 (0.79–4.30)	0.155	…	…
Pitt bacteremia score ≥4	2.77 (1.10–6.93)	0.030	1.28 (0.40–4.10)	0.683
SOFA score ≥6	2.30 (0.91–5.79)	0.078	…	…
Age ≥65 year	1.54 (0.66–3.60)	0.313	…	…
BMI <21 kg/m^2^	**2.43 (1.04–5.71)**	**0.041**	**4.58 (1.54–13.7)**	**0.006**

Abbreviations: aOR, adjusted odds ratio BMI, body mass index; CI, confidence interval; cOR, crude odds ratio; DD, disk diffusion; MDR, multidrug-resistant; SOFA, sequential organ failure assessment. Bold values indicate statistical significance.

## Data Availability

All data are included in the main manuscript and the Appendix A.

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
