# Peer review of "Direct Disk Diffusion Testing and Antimicrobial Stewardship for Gram-Negative Bacteremia in the Context of High Multidrug Resistance"

_antibiotics, 2025, doi:10.3390/antibiotics14070726_

Round 1
Reviewer 1 Report
Comments and Suggestions for Authors
- The topic of the manuscript is addressed to the main issue in the clinical microbiology. it will be better if the author extends the introduction part, in addition to give a comparison of recent published results.
- Authors are excluded some patients from the intervention group due to physician non-adherence. It will be better if the authors give more detailed description/explanation of the exclusion criteria and how these can influence the comparability of groups.
- The authors did not mention differences in mortality in the discussion and limits sections.
- Please, double check reference style according to the Antibiotics guidelines. In some cases, it did not fit.
- The Figure 1 (flowchart) is good to describe the main idea of the paper, but it will be better to improve graphically to improve visuality and clarity.
- The results describe with median (IQR) and mean (SD) values of similar parameters. Please standardize reporting within a manuscript.
- The "optimal therapy" and "ineffective therapy" relies on rather subjective parameters. Please provide a clearer definition or score criteria.
Author Response
- The topic of the manuscript is addressed to the main issue in the clinical microbiology. it will be better if the author extends the introduction part, in addition to give a comparison of recent published results.
Ans: Thank you for your valuable suggestion. We have revised and expanded the introduction section by including the names of commonly used techniques for rapid antimicrobial susceptibility testing (RAST) to better highlight their relevance to the main issue in clinical microbiology, as you recommended. We also compared with recent publications to provide further context (Line 72–75, 82–84).
- Authors are excluded some patients from the intervention group due to physician non-adherence. It will be better if the authors give more detailed description/explanation of the exclusion criteria and how these can influence the comparability of groups.
Ans: Thank you for your insightful comment. We have clarified the inclusion and exclusion criteria in Section 4.2 and expanded the discussion to address how physician non-adherence may have influenced the study outcomes, as you suggested (Lines 283–289).
- The authors did not mention differences in mortality in the discussion and limits sections.
Ans: Thank you for your kind comment. We have revised the manuscript to include additional discussion of the differences in mortality, as well as their implications, in both the Discussion and Limitations sections.
- Please, double check reference style according to the Antibiotics guidelines. In some cases, it did not fit.
Ans: Thank you for your kind comment. We have carefully reviewed and revised the references to ensure they conform to the Antibiotics journal’s guidelines.
- The Figure 1 (flowchart) is good to describe the main idea of the paper, but it will be better to improve graphically to improve visuality and clarity.
Ans: Thank you for your helpful suggestion. We have revised Figure 1 to improve its visual clarity and graphical presentation. The updated version includes clearer alignment, consistent formatting, and improved layout to enhance readability and facilitate understanding of the study flow. We believe these changes better convey the enrollment process and group assignments.
- The results describe with median (IQR) and mean (SD) values of similar parameters. Please standardize reporting within a manuscript.
Ans: Thank you for your thoughtful comment. We have standardized the presentation of data throughout the manuscript and clarified our approach in the Methods section. Specifically, we added the following sentence “Mean ± SD was used for normally distributed variables, while median (IQR) was reported for skewed data or variables with outliers, as the median better reflects central tendency in such cases.” This addition aims to improve transparency and consistency in the reporting of our results.
- The "optimal therapy" and "ineffective therapy" relies on rather subjective parameters. Please provide a clearer definition or score criteria.
Ans: Thank you for your insightful comment. We agree that the definitions of "optimal therapy" and "ineffective therapy" require clear and objective criteria. In our study, we defined optimal therapy as “the narrowest-spectrum antibiotic to which the causative pathogen was susceptible. The selected agent must have appropriate pharmacokinetics and pharmacodynamics (PK/PD) for the primary infection site, or there must be no narrower-spectrum alternative available.”
Conversely, ineffective therapy was defined as the use of an antibiotic to which the pathogen was resistant or had intermediate susceptibility (excluding colistin, which lacks established susceptibility breakpoints). This category also included cases where patients received antibiotics inappropriate for the infection site in terms of PK/PD or failed to receive any effective agent, despite the availability of a more suitable alternative.
These definitions were adapted from Kim et al. (J Med Microbiol. 2018, 67, 325–331), which evaluated a rapid diagnostic test similar to ours. We have now clarified this in Section 4.5 of the manuscript.
Reviewer 2 Report
Comments and Suggestions for Authors
There are several points that are needed prior to other next process. Here are my comments for this manuscript:
Abstract
- The abstract writing does not seem smooth. Some sentences should be modified, edited, or deleted.
- In general, "Gram" is used more than "gram."
- Please modify the keywords. Five keywords are enough. Importantly, short words are better than phrases.
- Please use the same space between each line along to the whole manuscript.
Introduction
- Please add the information about the empirical antibiotic use for GNBSI in your studied area.
- Please give the examples of international guidelines that you mentioned (Line 51).
- Please add information about direct disk diffusion and RAST, particularly in terms of the advantages and disadvantages of each technique.
- There is no objective for this study mentioned in the introduction. Please add an objective of this study at the end of introduction.
Results
- The results can be improved to be smoother. Many of the explanations are overly short and lack conjunctions, which can lead to misunderstandings; therefore, incorporating some conjunctions would be beneficial. Please revise.
- This means 60 years for male or for overall. (Lines 77-78).
- What is the objective of the different testing among comorbidities? Why does the author statically analyze this factor? Please explain.
- Because of the lack of objective mentioned in the introduction, some presented results were written in a different way, and this cannot scope the main/important finding related to the objective.
- Captions of figures/tables are very short. The author should add some explanation related to the presented figure or use the longer phrase that represents the figures/tables overall.
- Please check the p-value along with the manuscript. The author uses 0.xxx at certain points (mostly in the text), but the table uses .xxx instead.
- Table 1: Why do some factors use mean/SD while some factors use median/IQR? Please clarify. The reason that the author mentioned is that depending on normality might not cover for all. Please revise and add the details for each selection in 4.6 Statistical analysis too.
- Mostly, the explanation of results in this form is descriptive statistics and lack of in-depth analysis, even if the author mentioned that you use inferential statistics like the independent t-test/Mann Whitney U test. Please explain the findings from your inferential statistical analysis more than that information description.
- Tables: Please observe and edit the table’s style of the Antibiotics.
- Figure 2: Please modify. The authors should add Y-axis’s title. Delete lines in the figure. Importantly, figure captions should be placed under the figure.
Materials and methods
- Please describe additionally the different characteristics between the standard care phase and the intervention phase.
- What is routine clinical management without RAST-guided antimicrobial interventions? Please describe and clarify.
- What is “targeted antimicrobial therapy”? Please explain additional information.
- Please rewrite this sentence. “Patients were excluded if they survived for less than 24 hours after blood culture collection.” This might lead to misunderstanding that the patients died/survived.
- Please provide the program for sample size calculation and the setting that the author uses for calculation.
- The author mentioned that 68 patients per group is the minimum required to achieve 80% statistical power. Why did you include 73 patients for the intervention phase group?
- Did the author isolate bacteria before antibiotic susceptibility testing? It seems the author did not identify before, or the subsection of bacterial isolation and identification is lacking in the manuscript. Please recheck.
- Please add some information about bacterial culture.
- Please check the proper incubation time of the disk diffusion method following the CLSI guidelines. For interpretation using CLSI, the author should at least identify the genus of bacteria prior to susceptibility testing. Please clarify how you obtain bacterial identification results.
- Please specify the version of CLSI that the author uses for interpretation of susceptibility.
- Please include a subsection on ethical considerations in the materials and methods section. This study involved human participants; therefore, the research ethical approval number or any other relevant evidence should be included.
Discussion
- Please add more details that why RAST-driven interventions improve antimicrobial optimization, but this intervention did not always translate into mortality benefits.
- What is “AMS”? Please explain more.
- The authors list several limitations from this study. Thus, please give recommendations or suggestions for further study.
Conclusion
- Please include the suggestions for further study depending on your recent findings to this part also (only concise sentence than in discussion).
Author Response
Abstract
- The abstract writing does not seem smooth. Some sentences should be modified, edited, or deleted.
Ans: Thank you for your valuable suggestion. We have revised the abstract to improve its clarity, flow, and overall readability, as recommended.
- In general, "Gram" is used more than "gram."
Ans: We replaced "gram." With "Gram" per your suggestion.
- Please modify the keywords. Five keywords are enough. Importantly, short words are better than phrases.
Ans: Thank you for your valuable suggestion. We have revised the keywords to: disk diffusion, rapid diagnostics, antimicrobial stewardship, bacteremia, Gram-negative. To meet the five-keyword limit, we have removed blood culture from the list.
- Please use the same space between each line along to the whole manuscript.
Ans: Thank you for your comment. We used Microsoft Word to prepare the manuscript, and the line spacing was set to 1.0 throughout, including the abstract. We are not certain why the abstract appears to have wider spacing—it may be related to the formatting of the journal's template. We apologize that we were unable to adjust it further, but we believe this issue will be resolved during the publication process.
Introduction
- Please add the information about the empirical antibiotic use for GNBSI in your studied area.
Ans: Thank you for your valuable comment. In response, we have added information on the empirical antibiotic treatment practices for GNBSIs in our setting. These revisions have been incorporated into the revised manuscript (Lines 52–68).
- Please give the examples of international guidelines that you mentioned (Line 51).
Ans: Thank you for your helpful suggestion. We have revised the sentence to specify the guideline by stating: “Timely administration of appropriate antibiotics is essential in improving outcomes for patients with sepsis and septic shock, as emphasized in as emphasized in the 2021 Surviving Sepsis Campaign guidelines” This clarification has been incorporated into the revised manuscript.
- Please add information about direct disk diffusion and RAST, particularly in terms of the advantages and disadvantages of each technique.
Ans: Thank you for your helpful comment. We added a brief overview of commonly used RAST techniques. We also highlighted the advantages and limitations of DD testing, which is more feasible in our setting. These changes have been incorporated in the revised version (Lines 71–75, and 79–84).
- There is no objective for this study mentioned in the introduction. Please add an objective of this study at the end of introduction.
Ans: Thank you for your helpful suggestion. We have added the summarizing sentences — 'This study aimed to evaluate whether integrating direct DD with AMS could shorten the time to optimal antibiotic therapy in patients with GNBSI. A secondary objective was to determine whether this approach could also reduce in-hospital mortality.' (Lines 96–98) — in accordance with your recommendation.
Results
- The results can be improved to be smoother. Many of the explanations are overly short and lack conjunctions, which can lead to misunderstandings; therefore, incorporating some conjunctions would be beneficial. Please revise.
Ans: Thank you for your valuable comment. We have revised the Results section to improve the flow and clarity by expanding the explanations and incorporating appropriate conjunctions, as recommended.
- This means 60 years for male or for overall. (Lines 77-78).
Ans: Thank you for your helpful suggestion. We were referring to the overall study population and have revised the sentence to: 'The average age of patients was approximately 60 years, and the majority were male,' for clarity (Lines 102–103).
- What is the objective of the different testing among comorbidities? Why does the author statically analyze this factor? Please explain.
Ans: Thank you for your thoughtful comment. The analysis of comorbidities was conducted to describe the baseline characteristics of the study population. As the study was conducted in a tertiary care center, most patients had one or more comorbid conditions. Therefore, this analysis aimed to provide context about the patient population rather than to test a specific hypothesis related to comorbidities.
- Because of the lack of objective mentioned in the introduction, some presented results were written in a different way, and this cannot scope the main/important finding related to the objective.
Ans: Thank you for your valuable feedback. As suggested, we have added a paragraph stating the study objectives at the end of the Introduction section to help readers better understand and focus on the main findings.
- Captions of figures/tables are very short. The author should add some explanation related to the presented figure or use the longer phrase that represents the figures/tables overall.
Ans: Thank you for your valuable feedback. We corrected the captions of table and figure per your suggestion. We demonstrated the captions in the revised manuscript below.
- Figure 1. Flow diagram of patient enrollment and inclusion in the standard of care (SOC) and intervention groups.
- Figure 2. Distribution of Gram-negative bacilli species isolated from the blood cultures of all participants.
- Table 1. Baseline characteristics of patients in the SOC and intervention groups.
- Table 2. Clinical and laboratory parameters on the day of blood culture collection in the SOC and intervention groups.
- Table 3. Infection-related clinical information and pathogen profile in the SOC and intervention groups.
- Please check the p-value along with the manuscript. The author uses 0.xxx at certain points (mostly in the text), but the table uses .xxx instead.
Ans: Thank you for your thoughtful comment. We have revised the p-value format in the tables to 0.xxx to ensure consistency with the format used in the text.
- Table 1: Why do some factors use mean/SD while some factors use median/IQR? Please clarify. The reason that the author mentioned is that depending on normality might not cover for all. Please revise and add the details for each selection in 4.6 Statistical analysis too.
Ans: Thank you for your valuable comment. We have revised the Methods section (Lines 394–400) to provide a clearer explanation regarding the choice of summary statistics. Specifically, we now state:
"Continuous variables were analyzed using the independent t-test or the Mann-Whitney U test, depending on data distribution. Results are presented as mean with standard deviation (SD) or median with interquartile range (IQR). Mean ± SD was used for normally distributed data, while the median was preferred for data with outliers, as it is less affected by extreme values and better represents the central tendency in such cases."
This clarification has also been added in accordance with your suggestion to Section 4.7.
- Mostly, the explanation of results in this form is descriptive statistics and lack of in-depth analysis, even if the author mentioned that you use inferential statistics like the independent t-test/Mann Whitney U test. Please explain the findings from your inferential statistical analysis more than that information description.
Ans: Thank you for your valuable feedback. We acknowledge that much of our initial explanation focused on descriptive statistics, as it was important to clearly present the baseline characteristics of the study population. However, we agree that a more in-depth interpretation of the inferential statistical findings is necessary. In response, we have revised Section 2.8 by including 95% confidence intervals and providing a more detailed explanation of the results, as you suggested. Additionally, inferential statistical analyses were applied in Sections 2.5 to 2.8, and the results are presented in Tables 4 and 5 to compare clinical outcomes and identify factors associated with mortality.
- Tables: Please observe and edit the table’s style of the Antibiotics.
Ans: We edited the table style to match with the journal per your suggestion.
- Figure 2: Please modify. The authors should add Y-axis’s title. Delete lines in the figure. Importantly, figure captions should be placed under the figure.
Ans: We adjusted the figure per your suggestion.
Materials and methods
- Please describe additionally the different characteristics between the standard care phase and the intervention phase.
Ans: Thank you for your insightful comment. In response, we have added a description of the differing characteristics between the standard care and intervention phases in the Materials and methods section (Lines 300–312), as you suggested.
- What is routine clinical management without RAST-guided antimicrobial interventions? Please describe and clarify.
Ans: Thank you for your valuable comment. As suggested, we have clarified the details of routine clinical management without RAST-guided antimicrobial interventions in the revised manuscript. (Lines 300–308)
- What is “targeted antimicrobial therapy”? Please explain additional information.
Ans: Thank you for your helpful comment. We have removed the term “targeted antimicrobial therapy” from the manuscript, as it may be ambiguous and does not accurately reflect our intended meaning.
- Please rewrite this sentence. “Patients were excluded if they survived for less than 24 hours after blood culture collection.” This might lead to misunderstanding that the patients died/survived.
Ans: Thank you for your valuable feedback. We agree that the original wording may cause confusion. To improve clarity, we have revised the sentence as follows:
“Patients were excluded if they died within 24 hours after blood culture collection.”
- Please provide the program for sample size calculation and the setting that the author uses for calculation.
Ans: Thank you for your valuable feedback. We provided the program for sample size calculation and the setting that we used for calculation in the section 4.2, as you suggested.
- The author mentioned that 68 patients per group is the minimum required to achieve 80% statistical power. Why did you include 73 patients for the intervention phase group?
Ans: Thank you for your thoughtful question. In practice, positive blood culture cases did not occur at a consistent rate, and on some days, multiple eligible patients were identified simultaneously, resulting in a slight over-recruitment. Additionally, we intentionally included a few additional patients to account for potential exclusions, particularly in cases where the primary physician declined participation. We believe this small excess does not impact the study’s statistical power or compromise the validity of the results.
- Did the author isolate bacteria before antibiotic susceptibility testing? It seems the author did not identify before, or the subsection of bacterial isolation and identification is lacking in the manuscript. Please recheck.
Ans: Thank you for your thoughtful comment. We have added a description of the bacterial isolation, identification, and susceptibility testing process to Section 4.1 of the manuscript for clarification.
- Please add some information about bacterial culture.
Ans: Thank you for your thoughtful question. We added the information about bacterial culture in the section 4.1
- Please check the proper incubation time of the disk diffusion method following the CLSI guidelines. For interpretation using CLSI, the author should at least identify the genus of bacteria prior to susceptibility testing. Please clarify how you obtain bacterial identification results.
Thank you for your valuable comment. We have added details about the bacterial culture process in Section 4.1 to clarify how susceptibility results were obtained. In the intervention group, the direct disk diffusion test was performed as soon as possible after the positive blood culture signal. However, reporting of the susceptibility results to the infectious disease physician was done only after bacterial identification by MALDI-TOF MS.
- Please specify the version of CLSI that the author uses for interpretation of susceptibility.
Ans: Thank you for your thoughtful comment. The susceptibility results during the study were interpreted using the CLSI M100 32nd edition, as stated in Section 4.3. However, for the assessment of the accuracy of direct disk diffusion (Section 2.4, Table A2), we retrospectively applied the most recent CLSI guidelines (35th edition). This has been clearly noted in the manuscript.
- Please include a subsection on ethical considerations in the materials and methods section. This study involved human participants; therefore, the research ethical approval number or any other relevant evidence should be included.
Ans: Thank you for your thoughtful comment. We included a subsection on ethical considerations in the materials and methods, as you suggested.
Discussion
- Please add more details that why RAST-driven interventions improve antimicrobial optimization, but this intervention did not always translate into mortality benefits.
- What is “AMS”? Please explain more.
Ans: (for Q1 and Q2): Thank you for your insightful comments. We have revised the discussion to further clarify why RAST-driven interventions improve antimicrobial optimization but do not always lead to mortality benefits. We expanded on the multifactorial nature of sepsis outcomes, the effectiveness of initial empirical therapy in many patients, and the limitations of a narrow AMS approach. We also added a definition and explanation of AMS, emphasizing how its scope may affect patient outcomes. These revisions can be found in the Discussion section (Lines 249–258).
- The authors list several limitations from this study. Thus, please give recommendations or suggestions for further study.
Ans: Thank you for your thoughtful comment. We fully agree with your suggestion and have revised the final sentence of the discussion section to include a recommendation for future research: “Future multicenter randomized controlled trials with larger sample sizes, calculated based on mortality rates, are needed to more definitively assess the clinical impact of RAST-integrated AMS programs, including their effects on long-term outcomes, cost-effectiveness, and resistance patterns” (Lines 289–292).
Conclusion
- Please include the suggestions for further study depending on your recent findings to this part also (only concise sentence than in discussion).
Ans: Thank you for your suggestion. We have revised the conclusion section to include a concise recommendation for further research, as suggested.
Reviewer 3 Report
Comments and Suggestions for Authors
This article summarizes authors’ findings towards the shortening of optimal antibiotic therapy by combining disk diffusion technique with antimicrobial stewardship approaches. As the importance of finding effective antimicrobial therapies rises day by day, these findings become more valuable to increasing number of researchers. There are several minor issues that need to be addressed before the manuscript can be accepted for publication.
- Both the “standard care” and the “intervention” regimes must be detailed in the Materials and Methods section. Both are explained with referencing to the “standard treatment guidelines and the healthcare entitlements available in Thailand”, which is not quite clear to most of the readers. Similarities and differences must be explained. Guidelines for the antibiotic adjustments must be explained in more detail and highlighted.
- The manuscript refers to: “After incubation, the diameters of the inhibition zones were measured, and the results were interpreted according to the most recent CLSI guidelines available at the time of the study [26].” How will changes in the CLSI guidelines affect future evaluations of similar studies?
- The limitations of the study has already been mentioned in the Conclusion remarks of the manuscript. Detailed suggestions to overcome these limitations would be helpful for future research on the antimicrobial stewardship approaches.
Author Response
Comments and Suggestions for Authors
This article summarizes authors’ findings towards the shortening of optimal antibiotic therapy by combining disk diffusion technique with antimicrobial stewardship approaches. As the importance of finding effective antimicrobial therapies rises day by day, these findings become more valuable to increasing number of researchers. There are several minor issues that need to be addressed before the manuscript can be accepted for publication.
- Both the “standard care” and the “intervention” regimes must be detailed in the Materials and Methods section. Both are explained with referencing to the “standard treatment guidelines and the healthcare entitlements available in Thailand”, which is not quite clear to most of the readers. Similarities and differences must be explained. Guidelines for the antibiotic adjustments must be explained in more detail and highlighted.
Ans: Thank you for your helpful suggestion. In response, we have added more detailed descriptions of both the standard of care and the intervention group in Section 4.1, as recommended. The phrase “standard treatment guidelines and the healthcare entitlements available in Thailand,” which was included in the Introduction based on another reviewer’s suggestion, has also been clarified to improve understanding for international readers. Additionally, we have provided examples of antibiotic adjustments based on specific pathogens and commonly used national referral guidelines to better illustrate the approach used in our setting in section 4.4.
- The manuscript refers to: “After incubation, the diameters of the inhibition zones were measured, and the results were interpreted according to the most recent CLSI guidelines available at the time of the study [26].” How will changes in the CLSI guidelines affect future evaluations of similar studies?
Ans: Thank you for your thoughtful question. We agree that updates to CLSI guidelines may influence future evaluations of similar studies, especially if changes are made to the interpretive breakpoints for specific antimicrobials. However, in our experience, most CLSI updates related to direct disk diffusion (DD) testing involve the addition of new agents or methods, rather than major revisions of existing breakpoints. For example, the CLSI 2025 edition introduced breakpoints for agents such as cefepime, piperacillin/tazobactam, and ampicillin/sulbactam for direct disk diffusion, which were not present in earlier versions. In contrast, the breakpoints for commonly used drugs like ceftriaxone have remained stable between 2022 and 2025. While future studies may adopt newer breakpoints, we believe our results remain valid and comparable, particularly for the core antimicrobials assessed. To support cross-study comparisons, we have clearly stated the CLSI version used during the study period.
- The limitations of the study has already been mentioned in the Conclusion remarks of the manuscript. Detailed suggestions to overcome these limitations would be helpful for future research on the antimicrobial stewardship approaches.
Ans: Thank you for your suggestion. We have revised the conclusion to ensure that the impact of our findings is presented appropriately and not overstated, in light of the study’s limitations.
Reviewer 4 Report
Comments and Suggestions for Authors
Line 50-63. For clarity improvement, try to avoid combining multiple ideas (timing, accuracy, and cost) into a long sentence. Otherwise, the justification for the low-cost alternative is meaningful, especially by commenting on RAST barriers.
Line 73: A short summarizing sentence on the hypothesis of the study should be in order.
Line 75-87: The significant difference in WBC (P = 0.016) should be discussed as a potential confounder.
Lines 124-130: If available, report the confidence intervals for error rates.
Table 4: Move it closer to its narrative for easier readability.
Line 155-164: You may consider stating the limitation of the small size here as well, not only in the conclusion section.
Lines 218–228: There is strong support for the clinical utility of diagnostic testing. You may emphasize the zero very major error rate as a key safety finding.
Line 238: You may introduce a phrase to clarify how primary physician's refusal may introduce bias. Otherwise, a well-written limitations section.
Conclusion section: Given the limitations, try not to overestimate the impact.
Lines 265-280: Section regarding DD testing procedure - Consider adding justification for including off-label agents like sitafloxacin.
Author Response
Comments and Suggestions for Authors
Line 50-63. For clarity improvement, try to avoid combining multiple ideas (timing, accuracy, and cost) into a long sentence. Otherwise, the justification for the low-cost alternative is meaningful, especially by commenting on RAST barriers.
Ans: Thank you for your valuable comment. We have clarified the message by revising the sentences (Lines 79–84) in accordance with your suggestion.
Line 73: A short summarizing sentence on the hypothesis of the study should be in order.
Ans: Thank you for your helpful suggestion. We have added the summarizing sentences — 'This study aimed to evaluate whether integrating direct DD with AMS could shorten the time to optimal antibiotic therapy in patients with GNBSI. A secondary objective was to determine whether this approach could also reduce in-hospital mortality.' (Lines 96–98) — in accordance with your recommendation.
Line 75-87: The significant difference in WBC (P = 0.016) should be discussed as a potential confounder.
Ans: Thank you for your thoughtful comment. We have added a discussion of the baseline WBC difference as a potential confounder in the Discussion section (Lines 276–278).
Lines 124-130: If available, report the confidence intervals for error rates.
Ans: Thank you for your suggestion. We apologize that we are unable to provide confidence intervals, as the presenting number were calculated from a single dataset, which does not allow for estimation of variability.
Table 4: Move it closer to its narrative for easier readability.
Ans: We change the position of Table 4 in accordance with your suggestion.
Line 155-164: You may consider stating the limitation of the small size here as well, not only in the conclusion section.
Ans: Thank you for your thoughtful suggestion. In response, we have added the small sample size as a limitation in the Discussion section, as recommended.
Lines 218–228: There is strong support for the clinical utility of diagnostic testing. You may emphasize the zero very major error rate as a key safety finding.
Ans: Thank you for your valuable comment. We agree that the absence of very major errors (VME) is an important finding that underscores the safety of the method. Accordingly, we have revised the discussion section (Lines 263–265) to include the sentence: “The absence of VME indicates that no patients were at risk of receiving ineffective antibiotics, highlighting the safety of using direct DD for antimicrobial adjustment.” as suggested.
Line 238: You may introduce a phrase to clarify how primary physician's refusal may introduce bias. Otherwise, a well-written limitations section.
Ans: Thank you for your valuable comment. We have clarified how the primary physicians’ refusal may have introduced bias by adding the following sentence to the discussion section: “Most of these were critically ill patients, for whom physicians were hesitant to de-escalate antibiotics based on preliminary results. Including these cases might have influenced the study outcomes, potentially showing a greater reduction in treatment costs.” (Lines 286–289), as you suggested.
Conclusion section: Given the limitations, try not to overestimate the impact.
Ans: Thank you for your suggestion. We have revised the conclusion to ensure that the impact of our findings is presented appropriately and not overstated, in light of the study’s limitations.
Lines 265-280: Section regarding DD testing procedure - Consider adding justification for including off-label agents like sitafloxacin.
Ans: Thank you for your comment. We have added the sentences (Lines 344–347) to justify the inclusion of off-label agents like sitafloxacin. In Thailand, sitafloxacin is occasionally used as an oral step-down option for XDR infections, especially when resistance to co-trimoxazole and other fluoroquinolones is common. Its inclusion in DD testing supports early, locally relevant treatment decisions.
Round 2
Reviewer 1 Report
Comments and Suggestions for Authors
Manuscript can accepted for publication.
Author Response
Thank you very much for your kind recommendation. We are grateful for your valuable comments and are pleased to hear that the manuscript is suitable for publication.
Reviewer 2 Report
Comments and Suggestions for Authors
E. coli and Klebsiella pneumoniae are the predominant bacteria in this study. Could you please discuss the most possible route of transmission of theses pathogens, and how you finding can improve for the route if these mibrobes?
Author Response
Comment: E. coli and Klebsiella pneumoniae are the predominant bacteria in this study. Could you please discuss the most possible route of transmission of theses pathogens, and how you finding can improve for the route if these mibrobes?
Response: Thank you for your valuable comment. We have added the sentence Both of which commonly reside in the gastrointestinal tract and correlate with the primary sources of infection in these patients." (Lines 131–132). to show the likely transmission routes of the identified pathogens and how our findings may help clarify these pathways.
Reviewer 3 Report
Comments and Suggestions for Authors
All comments raised by this Reviewer are addressed.
Author Response
Thank you for your thoughtful review. We appreciate your time and are glad that all of your comments have been satisfactorily addressed.